# A 12-mer Peptide of Tag7 (PGLYRP1) Forms a Cytotoxic Complex with Hsp70 and Inhibits TNF-Alpha Induced Cell Death

**DOI:** 10.3390/cells9020488

**Published:** 2020-02-20

**Authors:** Elena A. Romanova, Tatiana N. Sharapova, Georgii B. Telegin, Alexei N. Minakov, Alexander S. Chernov, Olga K. Ivanova, Maxim L. Bychkov, Lidia P. Sashchenko, Denis V. Yashin

**Affiliations:** 1Laboratory of Molecular immunogenetics of cancer, Institute of gene biology RAS, Vavilova 34/5, 111394 Moscow, Russia; elrom4@rambler.ru (E.A.R.); sharapovatat.nik@gmail.com (T.N.S.); olga.k.ivanova@gmail.com (O.K.I.); sashchenko@genebiology.ru (L.P.S.); 2Branch of Shemyakin and Ovchinnikov Institute of Bioorganic Chemistry of the Russian Academy of Sciences, Prospect Nauki 6, 142290 Pushchino, Russia; telegin@bibch.ru (G.B.T.); minakov@bibch.ru (A.N.M.); alexandrchernov1984@gmail.com (A.S.C.); 3Laboratory of bioengineering of neuromodulators and neuroreceptors, Shemyakin-Ovchinnikov Institute of Bioorganic Chemistry RAS, Miklucho-Maklaya str., 16/10, 117997 Moscow, Russia

**Keywords:** Hsp70, Tag7, TNFR1, apoptosis, necroptosis, rheumatoid arthritis, peptides

## Abstract

Investigation of interactions between a pro-inflammatory cytokine tumor necrosis factor (TNFα) and its receptor is required for the development of new treatments for autoimmune diseases associated with the adverse effects of TNFα. Earlier, we demonstrated that the innate immunity protein Tag7 (PGRP-S, PGLYRP1) can interact with the TNFα receptor, TNFR1, and block the transduction of apoptotic signals through this receptor. A complex formed between the Tag7 protein and the major heat shock protein Hsp70 can activate TNFR1 receptor and induce tumor cell death via either apoptotic or necroptotic pathway. In this study, we show that a 12-mer peptide, designated 17.1, which was derived from the Tag7 protein, can be regarded as a novel TNFα inhibitor, also is able to form a cytotoxic complex with the heat shock protein Hsp70. This finding demonstrates a new role for Hsp70 protein in the immune response. Also, this new inhibitory 17.1 peptide demonstrates an anti-inflammatory activity in the complete Freund’s adjuvant (CFA)-induced autoimmune arthritis model in laboratory mice. It appears that the 17.1 peptide could potentially be used as an anti-inflammatory agent.

## 1. Introduction

Cytokines play a crucial role in the development of the immune system, eliciting immune responses, and often preventing autoimmune reactions [1]. TNF (tumor necrosis factor) is one of the oldest known cytokines, that plays an important role in the mechanisms of inflammation and programmed cell death [2,3] TNF contributes to inflammatory processes by enhancing expression of the cell adhesion molecules. Furthermore, TNF can induce proliferation of the immune cells via activation of the NFκB transcription factor [4,5,6], thus playing an important role in immune defenses against different diseases [7]. The mechanisms of TNF signal transduction are well-known [8,9]. The interaction between TNF and its receptor, TNFR1, causes the assembly of the cytoplasmic protein complex responsible for the type of signal transduced inside the cell. A signal for induction of the NFκB transcription factor, leading to activation of immune cells, is generated in some cases, while apoptotic or necrotic cascades are activated in other cases. A number of autoimmune diseases are associated with an overexpression of TNF [10,11]. One of these diseases is rheumatoid arthritis, which is associated with the TNF-dependent inflammation of the joints [12]. Antibodies reducing the amount of soluble TNF capable to interact with its receptor are currently widely used to combat autoimmune diseases associated with the TNF overexpression [13]. Another promising strategy for reduction of inflammation in the autoimmune diseases is to develop agents preventing signal transduction through the TNF–TNFR1 system [14]. For example, we have recently demonstrated that innate immunity regulator Tag7 can bind to the TNFR1 protein, a specific TNF receptor [15].

The mammalian Tag7 protein (a.k.a., PGRP-S and PGLYRP1) was originally described by researchers from the Institute of gene biology RAS [16]. Later, it was demonstrated that its homolog plays a crucial role in the innate immunity in insects by interacting with the Toll-like receptors [17,18]. Unlike its homolog in the insects, whose role in the immune defense has been studied extensively, the role played by the mammalian Tag7 protein has been insufficiently investigated. Most researchers focus on the antibacterial role of this protein in immune response [19]. We have studied the antitumor effect of this protein and have demonstrated that the complex formed between Tag7 and Mts1 protein can cause migration of immune cells [20], and can induce activation of lymphocytes, pointing to the Tag7 antitumor activity [21]. Another important function of Tag7 protein is its interaction with the heat shock protein Hsp70. The resulting Tag7–Hsp70 complex interacts with the TNFα receptor, TNFR1, thus inducing apoptosis and necroptosis in tumor cells [15]. The Tag7 protein of the Tag7–Hsp70 complex binds to TNFR1, while Hsp70 is needed for the cytolytic signal transduction. The Tag7 protein can be used to disrupt interactions between the TNFR1 receptor and its ligands for a treatment of inflammation in autoimmune diseases associated with the TNF expression. However, the exact fragment of Tag7 which binds to the TNFR1 receptor and to the Hsp70 molecule is unknown.

The objective of this study was to identify the Tag7 fragment capable of interaction with the TNFR1 receptor and Hsp70, and to study its potential to induce tumor cell death (in a complex with Hsp70) as well as to inhibit the interaction between TNFR1 receptor and its ligands (the Tag7–Hsp70 complex and TNFα cytokine).

## 2. Materials and Methods

### 2.1. Cell Culture

L929 cells were cultured in complete grown medium (DMEM supplemented with 10% fetal calf serum, L-glutamine and antibiotics). Cells were incubated at 37 °C with 5% CO_2_. This cell line was obtained from the cell line collection of N. N. Blokhin National Medical Research Center of Oncology of the Ministry of Health of Russia. The HEK293 human embryonic kidney cell line was cultured in DMEM with 10% fetal calf serum, penicillin (100 U/mL), and streptomycin (100 mg/mL) in a humidified 5% CO_2_ atmosphere at 37 °C. The TNFR1 knockdown via siRNA in HEK293 was performed as described in [15].

### 2.2. Proteins and Antibodies

The cDNAs encoding recombinant human Tag7 (GenBank accession number NM_005091) was subcloned in pQE-31 and expressed in *E. coli* М15 (pREP4) (Qiagen, Hilden, Germany). Tag7 was purified as described earlier [22]. The cDNAs for the recombinant human 70 kDa heat shock protein 1A (Hsp70) and human TNFα (GenBank accession numbers: NM_005345 and NM_000594, respectively) were subcloned into pQE-31 and expressed in *E. coli* M15 (pREP4) (QIAGEN, Hilden, Germany). Hsp70 and TNFα were purified on a Ni-nitrilotriacetic acid agarose column (QIAGEN Hilden, Germany) according to the manufacturer’s instructions.

Rabbit polyclonal antibodies to murine TNFR1 and soluble sTNFR1 were procured from Sigma Aldrich (St. Louis, Missouri, USA).

### 2.3. Affinity Chromatography, Immunoadsorption, and Immunoblotting

We used our own electrophoresis technique to detect the low-molecular-weight peptide 17.1 (1.5 kD). A 16% polyacrylamide gel was run not to the full length of the glass, so that the low-molecular weight compounds did not run out of the gel. As the peptide marker, we used a synthesized pure peptide 17.1, placed in the adjacent lane (see Supplemental Information) The Hsp70, sTNFR1, and 17.1 peptide were conjugated to CNBr-activated Sepharose 4B (GE Healthcare Chicago, Illinois, USA) according to the manufacturer’s protocol. The Hsp70, sTNFR1, 17.1, and 17.0 peptides were adsorbed onto the respective Sepharose 4B column. The column was thoroughly washed with PBS (phosphate buffered saline)/0.5 М NaCl and PBS alone, and then eluted with 0.25 M triethylamine, pH 12. The eluted material was resolved by SDS-PAGE and blotted onto a nitrocellulose membrane. The biotinylated products were visualized by incubating the membrane with streptavidin-conjugated horseradish peroxidase (HRP) and then with an ECL Plus^®^ kit (GE Healthcare Chicago, Illinois, USA). To detect sTNFR1, the blot was incubated with the rabbit anti-TNFR1 antibodies (1:10,000) and a secondary HRP-conjugated anti-rabbit antibody (GE Healthcare Chicago, Illinois, USA; 1:40,000) and then developed with an ECL Plus^®^ kit.

### 2.4. Cytotoxicity Assays

Cytotoxicity was evaluated using Trypan blue staining as described previously [23]. Cytotoxicity was calculated as:(1)$$Cytotoxity=\frac{\left(St-Sp\right)}{\left(T-Sp\right)}\times 100\%$$
where *St* is the number of stained cells; *Sp*, spontaneously stained cells; *T*, total cells. The live and dead cells were counted under a microscope (no less than 100 cells per assay). Peptide fractions or 17.1 peptide were added 1 h before addition of TNFα or the Tag7–Hsp70 complex.

### 2.5. Mass Spectrometry

Matrix Assisted Laser Desorption/Ionization mass spectorometry (MALDI) analysis was performed in the Institute of Biomedical Chemistry (IBMC). Data analysis was carried out using MASCOT^®^ Peptide Mass Fingerprint software (Matrix Science, http://www.matrixscience.com/cgi/search_form.pl) and NCBInr database (Japan).

### 2.6. Peptides

Protein Tag7 was hydrolyzed at 37 °C for 3.5 h at a 1:10 trypsin/protein ratio (*w/w*) in 50 mM (NH_3_)HCO_3_ (рН 8.0). The hydrolysate was then separated on a Superdex Peptide column. Peptides were synthesized on an automated peptide synthesizer according to the Fmoc strategy, and HATU (Hexafluorophosphate Azabenzotriazole Tetramethyl Uronium) was used as a coupling agent. Amino acids were taken in an eightfold excess; condensation of each amino acid was conducted for 30 m. The *C*-terminal amino acids were attached to activated resin in the presence of DIPEA for 2 h. After synthesis, the protected peptidyl polymer was washed with diethyl ether, dried, and treated with a TFA/DTT/H_2_O/TIS mixture (150/4/3/0.5 wt. %) (15 mL of the mixture per g of peptidyl polymer) for 2 h. The solution was filtered; the raw peptide was precipitated with a tenfold volume of diethyl ether and allowed to stand at 4 °C for 8 h. The precipitated peptide was centrifuged, washed three times with diethyl ether, and dried under vacuum. The raw peptide was purified by HPLC on an YMC Actus Triart C18 10u 30 × 150 mm column in 5–55% acetonitrile gradient and lyophilized.

### 2.7. Confocal Microscopy

L929 cells were grown on glass coverslips and fixed with 3.7% formaldehyde (prepared in 0.9% NaCl at pH 7.4) for 20 min at 37 °C. Cells were rinsed 3 times with PBS, and then the samples were placed into blocking solution (1% BSA in PBS) for 20 min at room temperature. Primary antibodies were added and incubated for an hour at room temperature. After rinsing, secondary antibodies (fluorochrome-labeled) were added and incubated for 60 min at room temperature in the dark. After washing with PBS the coverslips were embedded in ProLong Gold (Invitrogen, Carlsbad, USA). Rabbit anti-TNFR1 antibodies were from Santa Cruz Biotechnology (sc-7895). Mouse anti-PGRP-S antibodies were from Imgenex. The secondary antibodies were goat anti-rabbit Alexa Fluor 546 (A11035) and donkey anti-mouse Alexa Fluor 488 (A21202) from Life Technologies.

### 2.8. Mice

Female ICR mice housed at the nursery of laboratory animals of the Branch of the Institute of Bioorganic Chemistry, Russian Academy of Sciences, under standard conditions were used in this study. All the experiments and manipulations performed were approved by the Institutional Animal Care and Use Committee (IACUC) (no. 654/18 from 16/07/18). Arthritis was induced by local periarticular single-dose injection of 40 µL of complete Freund’s adjuvant (CFA) into the left ankle joint of mice. Peptide 17.1 (120 µg/mouse) dissolved in 100 µL of normal saline was injected subcutaneously into the interscapular region 24 h after the induction of inflammation. All mice were euthanized 2–4, 6–10, or 21 days later; and histological study of ankle joint specimens was performed. 

### 2.9. Histological Studies

The specimens (the tarsus and the lower part of the leg) were fixed with 10% formalin (pH = 7.4), demineralized for 2 days in BioDec decalcifying solution (BIOVITRUM, Saint-Petersburg, Russia), dehydrated using 7 portions of 99.7% isopropanol (5 h each), impregnated in two portions of paraffin medium (Histomix^®^, two h each), and then were embedded into it. Sagittal cross-sections (3 µm thick) were prepared using a rotary microtome and stained with hematoxylin and eosin according to the routine protocol. The specimens were analyzed on an AxioScopeA1 microscope (CarlZeiss, Oberkochen, Germany); the images were recorded using a high-resolution MRc.5 digital camera (CarlZeiss, Oberkochen, Germany). The severity of various arthritis manifestations was assessed semi-quantitatively according to the scale described in article [24]. Score 0–3 was used to assess periarticular inflammation (the intensity of infiltration of white blood cells into soft tissues surrounding the joint), synovitis (infiltration of WBCs into the synovial membrane), synovial hyperplasia, articular cartilage damage, and destruction of bone tissue.

### 2.10. Statistical Analysis

An unpaired two-tailed Student’s *t* test was used to determine statistical significance (Section 3.1). One-way ANOVA Dunnet was used to determine statistical significance in Section 3.2 and Section 3.3. *P*-values of less than 0.05 were considered significant. Data were analyzed using GraphPad Prism8 software. (GraphPad Software San Diego, CA, USA)

## 3. Results

### 3.1. Tryptic Peptide Derived from Tag7 Protein Inhibits the Cytotoxic Activity of TNFα

During the first stage of our study, we identified the region of Tag7 molecule that was inhibiting the interaction between TNFα and its receptor. Tryptic hydrolysis of protein Tag7 was performed for this purpose. Tag7 was subjected to restricted proteolysis with trypsin to obtain 1–2 kDa peptides that were optimal for analysis and the subsequent amino acid synthesis. The proteolytic conditions were selected experimentally. Gel filtration chromatography of the products of 4 h hydrolysis on a Superdex Peptide column identified several fractions having a molecular weight of 1–3 kDa.

Next, we studied the ability of each fraction to inhibit the cytotoxic activity of TNFα. For this purpose, L929 cells were preincubated with peptides from each fraction, and the cytotoxic activity was evaluated after adding TNFα. Untreated cells were used as a control. Figure 1 demonstrates that among 16 analyzed fractions, only one fraction (number 11) contained a peptide that almost completely inhibited TNFα-induced cytolysis. Fraction 1 contained non-hydrolyzed Tag7. MALDI analysis was performed to identify the amino acid sequence of the 12-membered peptide within this fraction. It showed that this peptide (snyvlkghrdvqr) resides at the C-terminus of Tag7 molecule (amino acid residues 163–175).

### 3.2. Peptide 17.1 Binds to TNFRI Receptor and Inhibits Cytotoxic Activity of TNFα and the Tag7–Hsp70 Complex

Once the amino acid sequence of the inhibitory peptide had been elucidated, this peptide was chemically synthesized and purified. A control peptide exhibiting no inhibitory activity was also synthesized. The structure of this peptide matched that of the N-terminus of Tag7 (amino acid residues 78–87). During synthesis, the inhibitory and control peptides were designated as 17.1 and 17.0, respectively. 

One can see in Figure 2A that peptide 17.1 inhibited the cytotoxic activity of both TNFα and the Tag7–Hsp70 complex. Peptide 17.0 had no effect on activity of these cytotoxic proteins. Similar data were obtained when HEK293 cells were used to count the cytotoxic activity.

Evaluation of the dependence between the cytotoxic activity of these proteins and concentration of peptide 17.1 demonstrated that the inhibition was dose-dependent and reached half of its efficiency when the inhibitor was taken in a twofold excess with respect to TNFα and a fivefold excess with respect to the Tag7–Hsp70 complex (Figure 2B). The IС_50_ values were 0.2 and 0.5 nM, respectively, and were comparable to IC_50_ observed for the full-length Tag7 molecule, which also has an inhibitory effect on cytotoxicity of these proteins. (IC_50_ was 2.2 and 5.2 nM, respectively) [15]. The addition of increasing concentrations of TNFα to the cells preincubated with 17.1 peptide lead to a dose-dependent restoration of cytotoxic effect, while 10-fold excess of TNFα fully overcame the inhibitory effect of the 17.1 peptide (Figure 2C).

We demonstrated earlier that the full-length Tag7 molecule binds to TNFRI receptor and inhibits apoptosis caused by TNFα and the Tag7–Hsp70 complex by impeding the interaction between these proteins and receptors ligand binding domain. It appeared reasonable to assume that peptide 17.1 also inhibits cytotoxicity due to its interaction with TNFRI.

Next, we tested the binding of biotinylated peptides 17.1 and 17.0 to the soluble extracellular domain of TNF receptor (sTNFRI). The results shown in Figure 2D demonstrate that peptide 17.1 binds to sTNFRI immobilized on CNBr-activated sepharose. As one would expect, the control peptide 17.0 does not bind to sTNFRI.

Confocal microscopy studies verified that peptide 17.1 interacts with sTNFRI. One can see that peptide 17.1 colocalizes with sTNFRI on the cell surface (Figure 3A).

The replacement of peptide 17.1 from its complex with sTNFRI caused by 100-fold excess of TNFα (Figure 3B) demonstrates that both the peptide and the cytotoxic protein interact with the same region of the extracellular domain of the receptor.

Hence, we have demonstrated that peptide 17.1 interacts with the extracellular domain of TNFRI receptor (both immobilized on CNBr-activated Sepharose and present on the cell surface) and that this interplay may impede the binding of TNFα or the Tag7–Hsp70 complex to this domain of the receptor, thus inhibiting the cytotoxic activity of these proteins. Earlier, we demonstrated that the full-length molecule of Tag7 protein also binds to TNFRI and inhibits the cytotoxicity of TNFα or the Tag7–Hsp70 complex [15]. The results reported above give grounds for assuming that the domain responsible for binding to TNFRI resides in the C-terminus of Tag7 molecule and matches the amino acid sequence of peptide 17.1.

### 3.3. Peptide 17.1 Forms a Cytotoxic Complex 17.1–Hsp70

Earlier, when studying the mechanism of cytotoxic activity of the Tag7–Hsp70 complex, we demonstrated that only Tag7 can bind to TNFRI out of the two components of this protein complex. Furthermore, only Hsp70-bound Tag7 protein can induce transduction of the cytotoxic signal into the cells. In this study, it has been demonstrated that peptide 17.1 binds to TNFRI. It can be assumed that this peptide within its complex with Hsp70 can exhibit a cytotoxic effect on cells, in a similar manner to the full-length molecule. To test this assumption, we had to answer two questions: (1) whether peptide 17.1 can interact with Hsp70 and (2) whether this complex exhibits a cytotoxic activity.

First, we investigated binding between peptide 17.1 and Hsp70. Chromatographic study using a column packed with Hsp70 immobilized on sepharose followed by SDS-PAGE electrophoresis and immunoblotting demonstrated that peptide 17.1 binds not only to sTNFRI, but also to Hsp70. No interaction was detected between the control peptide 17.0 and Hsp70. Hence, the 12-membered peptide can bind to proteins having different spatial structures. (Figure 4A.)

Next, we tested whether the 17.1–Hsp70 complex exhibited cytotoxic activity. The results presented in Figure 4B demonstrate that the complex formed between peptide 17.1 and Hsp70 protein may cause lysis of target cells like the complex between the full-length Tag7 protein and Hsp70 does. The cytotoxic activity was inhibited after preincubation of cells in the presence of proteins interacting with TNFRI (anti-TNFRI antibodies and Tag7 protein). The same effect was determined when HEK293 cells were used to count the cytotoxic activity. HEK293 cells with the knocked down *tnfr1* expression, were resistant to the cytotoxic effect of 17.1-Hsp70 complex.

Preincubation of the cytotoxic 17.1–Hsp70 complex in the presence of sTNFRI at increasing concentration also reduced the cytotoxic activity of this complex (Figure 4C). These results infer that the cytotoxic complex under study can bind to sTNFRI in solutions, and this binding prevents its interaction with TNFRI receptor on the cell surface.

However, the efficiency of cytotoxic activity of the 17.1–Hsp70 complex is somewhat lower than that of TNFα and the full-length Tag7–Hsp70 complex. The maximum cytotoxic activity of the 17.1–Hsp70 complex is achieved at a concentration of 0.5 nM (Figure 4D), which is higher than the maximum concentrations of TNFα and Tag7–Hsp70 (0.05 and 0.1 nM, respectively, [15]).

As it was suggested earlier for the full-length Tag7–Hsp70 complex, it is fair to assume that cell death caused by the 17.1–Hsp70 protein is also a two-stage process where each component has its own function. Peptide 17.1 can ensure binding of the complex to the receptor. Hsp70 capable of oligomerization can be involved in modifying the structure of the cytoplasmic domain of the receptor. To verify this assumption, each component of this complex was added separately to the cells. The results of this experiment are shown in Figure 4E. First, the cells were incubated with peptide 17.1 (one can see that the peptide does not induce cell lysis, identically to the full-length protein) [15]. The cells were then thoroughly washed to remove any unbound peptide, and Hsp70 was added to them. The cytotoxic effect was observed already at an equimolar ratio between Hsp70 and the peptide. When Hsp70 was taken in a ten-fold excess, the cytotoxic activity of the complex formed by it on the cell surface was maximum. Hence, Hsp70 can interact with peptide 17.1 that has already been bound to TNFRI and induce cell lysis.

The results presented above demonstrate that the structural region of Tag7 protein responsible for binding with Hsp70 and the TNFRI receptor has been revealed. A cytotoxic complex capable of killing cells was formed in the former case, while in the latter case, TNFRI-mediated cell death was inhibited. Peptide 17.1 efficiently inhibits the cytotoxic activity of TNFα. Since TNFα is involved in induction of inflammation, it is fair to expect that peptide 17.1 would exhibit a protective effect in these processes. To test this assumption, we further studied the anti-inflammatory activity of peptide 17.1 using a model of autoimmune arthritis.

### 3.4. Peptide 17.1 Exhibits an Anti-Inflammatory Activity in the Model of CFA-Induced Arthritis

The inflammation was simulated by injecting CFA into the periarticular tissues of the mouse ankle joint. At the first stage after immunizing mice with CFA, various growth factors and cytokines (including TNFα) are released and exhibit a pathological effect on cells.

The activity of peptide 17.1 was compared to that of Norocarp drug that is conventionally used to treat arthritis. Biospecimens of murine joints were harvested during the acute (day 2–4 after induction of inflammation), intermediate (days 6–10 after induction of inflammation), and chronic phases (day 21 after induction of inflammation). During the histological analysis, a rating scale (with score ranging from 0 to 3) was used to assess periarticular inflammation (the intensity of infiltration of WBCs into soft tissues surrounding the joint), synovitis (infiltration of WBCs into the synovial membrane), synovial hyperplasia, articular cartilage damage, and destruction of bone tissue. The results are summarized in the Table 1.

Injection of CFA severely affected the periarticular tissues (periarthritis with exudation). One can see that the minimal synovitis severity during the intermediate phase was observed for mice treated with peptide 17.1 after CFA injection (score 1) compared to mice receiving Norocarp and normal saline (score 1.29 and 1.86, respectively). In the experiment with chronic inflammation, the severity of articular cartilage damage and destruction of bone tissue was minimal after injection of peptide 17.1 in mice with induced inflammation (score 0.5 and 0.75, respectively).

The aforementioned data demonstrate that peptide 17.1, which was studied using the model of induced arthritis in mice, can exhibit a protective effect on articular cartilage and bone tissues, as well as inhibit synovitis in mice with CFA-induced arthritis.

## 4. Discussion

Three significant observations were made in this study: (1) the 12-membered peptide 17.1 can be regarded as a novel inhibitor of TNFα; (2) peptide 17.1 forms a cytotoxic complex with heat shock protein Hsp70; and (3) peptide 17.1 can exhibit an anti-inflammatory effect in laboratory mice with CFA-induced autoimmune arthritis.

Earlier, we demonstrated that Tag7 protein binds to TNFR1 (the TNFα receptor) and can have two opposite functions during cytolysis of tumor cells [15]. When binding to TNFR1, it does not induce the cytotoxic signal in the cell but inhibits the cytotoxic effect of both TNFα and the Tag7–Hsp70 complex. In its complex with Hsp70, Tag7 protein causes cell death mediated by its interaction with TNFR1. Here, we identified a structural fragment of Tag7 that is responsible for these functions, both the inhibitory and cytotoxic ones.

Attention should be paid to the fact that the 12-membered peptide can interact with two proteins carrying different amino acid sequences and differing in their spatial structure. Furthermore, the results reported above demonstrate that Hsp70 can bind to peptide 17.1 that has already been bound to TNFR1 receptor on the cell surface, without preliminary formation of the cytotoxic complex. These results prove our earlier hypothesis that the cytotoxic effect of the Tag7–Hsp70 complex on tumor cells involves two stages [25]. At the first stage, Tag7 protein binds to the TNFR1 receptor without signal transduction into the cell. The second stage, at which the cytotoxic signal is actually induced, is totally dependent on activity of Hsp70 bound to Tag7. It is possible that this mechanism utilizes the oligomerizing ability of Hsp70, which causes trimerization of the receptor required to ensure the cytotoxic effect. Binding of Hsp70 to peptide 17.1 that is already bound to TNFR1 on the cell membrane gives grounds for assuming that these proteins interact with different peptide regions.

It has been known for a long time that along with having a major chaperone function, the multifunctional Hsp70 protein is involved in regulation of immune response and affects tumor cell growth. Endogenous Hsp70 typically protects tumor cells by forming stable complexes with proteins involved in apoptotic signaling pathways, including JNK [26], AIF [27], granzyme B [28], and caspase 3 [29]. Hsp70 present on the cell membrane of tumor cells and makes them more susceptible to NK cells and cytotoxic T lymphocytes (CTL) [30,31]. Hsp70 released from tumor cells can act as a danger sign and elicit activation of cells responsible for immune response generation [32]. The findings reported above give grounds for assuming that Hsp70 has one more function. Extracellularly secreted Hsp70 can cancel the inhibition of cytotoxic activity induced via the TNFR1 receptor. By interacting with Tag7 protein already bound to TNFR1 on the cell surface, Hsp70 can convert the inactive Tag7–TNFR1 complex to the cytotoxic Tag7–Hsp70–TNFR1 complex.

As mentioned above, peptide 17.1 can be regarded as a novel inhibitor of TNFα activity. Therefore, we assumed that this peptide can inhibit inflammation in mouse models. We have simulated induced arthritis in laboratory mice using CFA, which induces the release of TNFα into tissues, to demonstrate that peptide 17.1 exhibits a protective effect on articular cartilage and bone tissues. Peptide 17.1 can turn out to be a promising medicinal agent capable of inhibiting inflammation.

## Figures and Tables

**Figure 1 cells-09-00488-f001:**
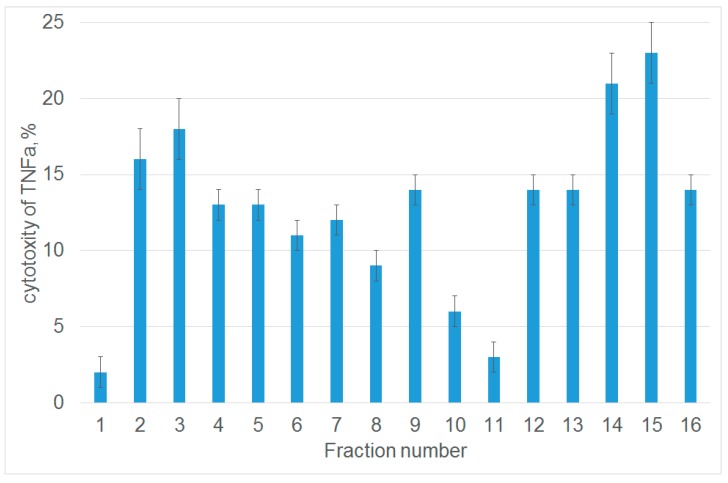
The peptides obtained after tryptic hydrolysis of Tag7 inhibit the cytotoxic activity of TNFα. The cytotoxic activity of TNFα (10^−9^ M) after 1 h preincubation of L929 cells in the presence of peptides derived from protein Tag7. A fraction of proteins eluted from the column and dissolved in DMEM medium was added to L929 cells; TNFα was added 1 h later. Cytotoxicity was measured after incubation for 20 h. Data are presented as the means ± SEM from three independent cytotoxic assays.

**Figure 2 cells-09-00488-f002:**
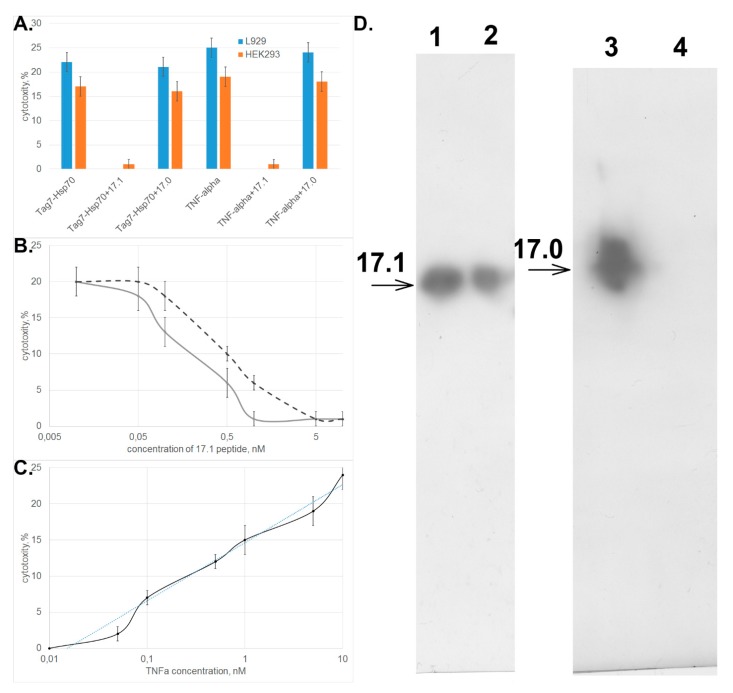
Peptide 17.1 can inhibit cytotoxicity of tumor necrosis factor (TNFα) and the Tag7–Hsp70 complex. (**A**) Inhibition of cytotoxic activity of the Tag7–Hsp70 complex (10^−10^ M) and TNFα (5 × 10^−11^ M) in the presence of peptides 17.1 and 17.0 (10^−9^ M) on L929 cells or HEK293 cells. (**B**) Cytotoxicity after 20 h incubation of TNFα (5 × 10^−11^ M, solid curve) and the Tag7–Hsp70 complex Hsp70 (10^−10^ M, dashed curve) after preincubation of L929 cells in the presence of peptide 17.1 at increasing concentration. (**C**) Cytotoxicity after 20 h incubation of increasing concentrations of TNFα after preincubation of L929 cells in the presence of peptide 17.1 (10^−9^ M). All data are presented as mean ± SEM for at least three independent replicates. (**D**) Binding of peptides 17.1 and 17.0 to sTNFR1. (2, 3) control for peptides 17.1 and 17.0, respectively. (1, 4) elution of 17.1 and 17.0, respectively, from the column packed with immobilized sTNFR1. Anti-Tag7 rabbit antibodies, with anti-rabbit antibodies conjugated to peroxidase added subsequently, were used for staining.

**Figure 3 cells-09-00488-f003:**
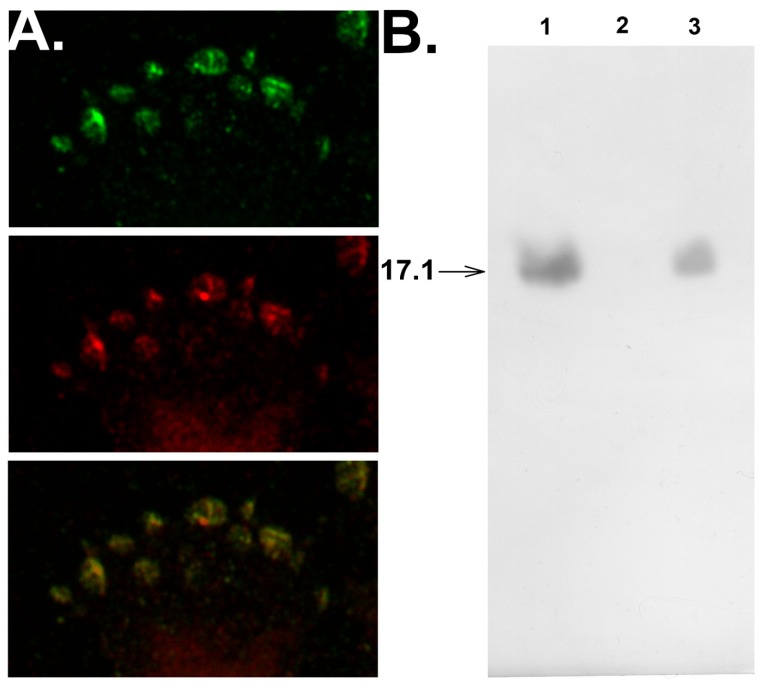
Peptide 17.1 binds to TNFR1 on the cell surface and competes for binding with TNFR1 in solution. (**A**) Confocal photograph of the surface of L929 cells after incubation with 17.1 peptide for 30 min, stained with anti-Tag7 antibodies (green), anti-TNFR1 antibodies (red), and image superposition. (**B**) Western blot analysis of 17.1 peptide (1), applied to the sTNFR1 conjugated Sepharose column and then excessively washed with PBS (2) and with TNFα protein (100× excess) (3).

**Figure 4 cells-09-00488-f004:**
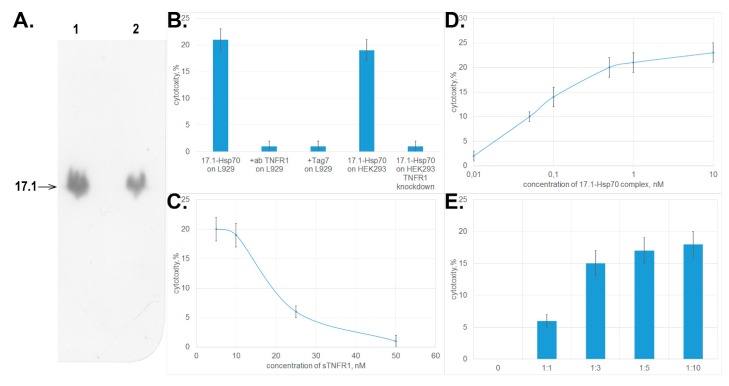
The complex formed between peptide 17.1 and Hsp70 causes tumor cell death. (**A**) Western blot showing 17.1 peptide binding to Hsp70. (1) control 17.1 peptide, (2) eluate of 17.1 from the column packed with immobilized Hsp70. (**B**) Cytotoxic activity of the 17.1–Hsp70 complex (10^−9^ M) after preincubation of L929 cells with anti-TNFR1 antibodies (1:100) and Tag7 protein (10^−8^ M) and cytotoxic activity of the 17.1–Hsp70 complex (10^−9^ M) on HEK293 cells and HEK293 cells with TNFR1 knockdown. (**C**) Preincubation of L929 cells with sTNFR1 inhibits the cytotoxic activity of the 17.1–Hsp70 complex (10^−9^ M). (**D**) Cytotoxicty after 20 h incubation of the 17.1–Hsp70 complex with L929 cells. (**E**) An excessive amount of Hsp70 added to L929 cells preincubated in the presence of peptide 17.1 results in emergence of cytotoxic activity. All data are presented as mean ± SEM for at least three independent replicates.

**Table 1 cells-09-00488-t001:** Peptide 17.1 exhibits an anti-inflammatory protective effect in mice with complete Freund’s adjuvant (CFA)-induced arthritis.

Number of Days after Inflammation Was Induced	Study Groups	Periarticular Inflammation (Intensity of Infiltration of White Blood Cells into SoftTissues Surrounding the Joint, Score)	Synovitis (Infiltration of WBCs into the Synovial Membrane, Score)	Synovial Hyperplasia (Score)	Articular Cartilage Damage (Score)	Destruction of Bone Tissue (Score)
**2–4 days**	NS + NS	0	0	0	0.25	0
	CFA + NS	2	0.67	0	0.33	0.33
	CFA + peptide	2.56	0.89	0.33	0.78	0.11
	CFA + Norocarp	2.25	1.75	1	1.00	0.25
**6–10 days**	NS + NS	0	0	0	0.33	0
	CFA + NS	3	1.86	1.25	0.75	0.75
	CFA + peptide	2.33	1	0.67	0.50	1.33
	CFA + Norocarp	2.5	1.29	0.25	0.75	1.25
**21 days**	NS + NS	0	0	0	0.25	0
	CFA + NS	2.75	0.5	0.5	1	2
	CFA + peptide	2.75	0.75	0.5	0.5	0.75
	CFA + Norocarp	2.75	0	0	0.75	1.5

CFA—complete Freund’s adjuvant; NS—normal saline.

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
