# Peer review of "A 12-mer Peptide of Tag7 (PGLYRP1) Forms a Cytotoxic Complex with Hsp70 and Inhibits TNF-Alpha Induced Cell Death"

_cells, 2020, doi:10.3390/cells9020488_

Round 1

Reviewer 1 Report

Romanova et al. describe in their article “A 12-mer Peptide of Tag7 (PGLYRP1) Forms Cytotoxic Complex with Hsp70 and Inhibits TNF-alpha Induced Cell Death” that the Peptide of Tag7 that has the ability to interact with TNFR1 and thus can block cell death.

In the current study the authors claim that their peptide 17.1 that is derived from Tag7 inhibits TNFalpha and forms a cytosolic complex with HSP70 thus demonstrating a new role for HSP70 in the immune response field. Furthermore, they claim an anti-inflammatory activity of peptide 17.1 in a CFA-induced autoimmune arthritis in a mouse model. According to Romanova et al., the peptide 17.1 might serve as potential novel therapeutic agent to inhibit inflammation.

One general weakness of the study is that the in vitro cytotoxicity studies were only performed on one cell line, L929 and thus not confirmed on a second model.

Figure 3: Immunofluorescence analysis gives a hint of a co-localization, but to confirm the interaction, an immunoprecipitation is required.

It is thus surprising that the authors claim that peptide 17.1 is their best peptide, they claim in Figure 1 that only fraction 11 almost completely inhibit the cytolysis of TNFalpha, but whatever fraction 1 is – the effect is even better than the effect of fraction 11.

Mouse model, table 1: the authors claim that peptide 17.1 is better than Norocarp for treatment, but depending on the investigated parameter, sometimes peptide 17.1 shows better score, sometimes Norocarp 3, sometimes they show equivalent values – independent of the phase.

minor

In Figure 2 and 4, no error bars (authors write in the figure legends SEM) are shown, axis labeling is wrong (in English, decimal numbers have to be marked with “.” and not with “,”)

Figure 2 and 3 – molecular weight markers should be shown and labelled.

Author Response

One general weakness of the study is that the in vitro cytotoxicity studies were only performed on one cell line, L929 and thus not confirmed on a second model.

We have added new data obtained on a second cell line, HEK293, in which we have also performed knockdown of TNFR1 receptor. This data is now included in the Figures, the Results,, and the Discussion chapters, as well as in the Materials and methods.

Figure 3: Immunofluorescence analysis gives a hint of a co-localization, but to confirm the interaction, an immunoprecipitation is required.

We believe that in addition to immunofluorescence, a direct interaction of peptide 17.1 with the TNFR1 receptor, is supported by the results of several other experiments presented in our manuscript:

1) Pre-incubation of peptide 17.1 with sTNFR1,the soluble extracellular receptor region, in PBS has reduced the inhibitory activity of the peptide, indicating that soluble sTNFR1 can bind to 17.1 peptide and compete for its binding to TNFR1 on the cell membrane.

2) We have used affinity chromatography in native conditions, which we believe very closely recapitulates conditions used in conventional immunoprecipitation analyses. We have shown that peptide 17.1 specifically binds to sTNFR1 when passed through a column with sTNFR1 immobilized on Sepharose at pH=7.4. The resultant complex is quite stable and dissociates only when the pH of the eluting solution is significantly increased (Figure 2C) or when an excess of another ligand of this receptor (TNFa) is added (Figure 3B).

3) The data we have added about the disappearance of cytotoxicity of the 17.1-Hsp70 complex on a clone of HEK293 cells with a reduced TNFR1 expression via siRNA knockdown also indicates the need for interaction of the 17.1 peptide with TNFR1 on the cell surface during the induction of cell death.

It is thus surprising that the authors claim that peptide 17.1 is their best peptide, they claim in Figure 1 that only fraction 11 almost completely inhibit the cytolysis of TNFalpha, but whatever fraction 1 is – the effect is even better than the effect of fraction 11.

In fraction 1, it was the exclusion volume of the column that was eluted, i.e., the proteins and peptides which exceeded its limit in molecular weight. MALDI analysis showed that this fraction contained a certain amount of the non-hydrolyzed Tag7 protein. Since we previously demonstrated that the Tag7 protein itself is capable of inhibiting the TNF-dependent cytotoxic activity (JBC 2015), we assumed that the effect of cytotoxicity inhibition caused by 1st fraction is associated with the activity of the non-hydrolyzed Tag7 protein, and did not consider it further. This information we have added now to the Results Chapter, to clarify this issue, and we are grateful to the reviewer for pointing out, that this important caveat was not described clearly.

Mouse model, table 1: the authors claim that peptide 17.1 is better than Norocarp for treatment, but depending on the investigated parameter, sometimes peptide 17.1 shows better score, sometimes Norocarp 3, sometimes they show equivalent values – independent of the phase.

The processes that take place in the CFA-induced arthritis model in mice are quite complex. It is known that the main source of chronic inflammation, which then leads to the destruction of cartilage and bone tissue, is TNFa, which is secreted by monocytes into the synovial cavity. We hypothesized that a prospective therapy that interfered with the TNFa action and that could block interaction of this cytokine with its specific receptor would help to reduce the severity of the disease and the associated destruction of cartilage and bone tissue. The tested peptide 17.1 demonstrated an effect of reduction in the destruction of cartilage and bone in the long-term period, which we regarded as a confirmation of our assumptions. Inflammatory processes in the joints could be associated not only with the TNFa action alone, that is why the stronger effect of a specific therapy with Norocarp in suppressing the primary signs of inflammation was more pronounced. In this work, our task was to study a possibility of protective action of the 17.1 peptide in the mouse arthritis model, in comparison with a traditional non-steroidal anti-inflammatory drug.

minor

In Figure 2 and 4, no error bars (authors write in the figure legends SEM) are shown, axis labeling is wrong (in English, decimal numbers have to be marked with “.” and not with “,”)

We are grateful to the reviewer for careful reading of our work and pointing out inaccuracies. We have added data on statistic analysis to the Materials and Methods and the Figure legends and have corrected the figures according to the reviewer's comments.

Figure 2 and 3 – molecular weight markers should be shown and labelled.

We used our own electrophoresis technique to detect the low-molecular-weight peptide 17.1 (1.5 kD). A 16% PAG was run not to the full length of the glass, so that the low-molecular weight compounds did not run out of the gel. As the peptide marker, we used a synthesized pure peptide 17.1, placed in the adjacent lane. This marker is shown in the figure. The usual markers of molecular masses are shown in the Supplemental information. We have added information on the SDS electrophoresis methodology to the Materials and Methods chapter.

Reviewer 2 Report

The authors continue and extend their previous studies of the Tag7 protein and the Tag7-Hsp70 complex as a potential TNFα inhibitor. In the present study, the authors claim to have identified a 12-mer peptide, called 17.1, at the C-terminus of Tag 7 that could be used as an anti-inflammatory drug.

The cell line used for this study was the L929 line, which is sensitive to TNFα induced cells dead. The experiments showed that the 17.1 peptide protects cells from TNFα induced cells death and binds to HSP70 and TNFR1. The binding of the peptide to the receptor alone blocks the cytotoxic effect of the TNFR1 ligands, the complex formed between 17.1 and HSP70 has instead cytotoxic activity.

The study is generally interesting and presents a moderate advance. Unfortunately, some of the data presented in the study are not of high quality (e.g. western blot images). Moreover, there are several inconsistencies between the text and figures. The rationale behind some experiments is not so clear and appropriate controls are missing or not mentioned in the text. In addition, replicates of the experiments should be indicated in the figure legends and statistics should be added when possible.

Specific comments

The authors performed cytotoxicity tests by using trypan blue exclusion or MTT assays. They should specify which test was used in each figure legend. In addition, these tests measure the percentage of dead cells or the percentage of cell viability. In fact, the axes of the graphs should indicate the percentage of dead cells or cell viability instead of “cytotoxic activity”. Line 162-164. The authors analysed 20 fractions, however Fig. 1A shows only 16 fractions. If the remaining samples were excluded the reason should be included in the text. Figure 1A. What does the fraction number 1 represent? It has a lower percentage of dead cells than fraction 11. Which control was used? Have the data been normalized to untreated cells? How many times was the experiment repeated? Statistics should be added. Line 176-177. What is the rationale behind the nomenclature of the peptide? The fraction isolated was the number 11. The names 17.1 and 17.0 are somewhat confusing. Figures 2C and 3B. The quality of the western blots is not acceptable. Lines 210-212. Which concentration of TNFα did the authors use? If the increase in concentration of 17.1 peptide causes a decrease in cytotoxicity mediated by TNFα, is the opposite also true? It would be more informative to repeat the experiment in Figure 3B using an increasing concentration of TNFα. Lines 236-240. The authors claim that Hsp70 interacts with the 17.1 peptide, and this is supposedly shown by western blot in Figure 4A. However the figure does not show what the authors claim but it is referred to in lines 241-246. The authors need to show the western blot they are referring to. Table 1. In this experiment it would have been appropriate to measure the release of inflammatory cytokine such as IL-6 to corroborate the conclusions.

Author Response

Specific comments

The authors performed cytotoxicity tests by using trypan blue exclusion or MTT assays. They should specify which test was used in each figure legend. In addition, these tests measure the percentage of dead cells or the percentage of cell viability. In fact, the axes of the graphs should indicate the percentage of dead cells or cell viability instead of “cytotoxic activity”.

We are grateful to the reviewer for pointing out inaccuracies and inconsistencies in our manuscript. All the cytotoxic activity data were obtained in our ms by the trypan blue staining method. We used MTT test only for verification of several tripan blue experiments, and since the data were essentially the same, we do not include it in the ms. In the new version of ms we have removed all mentions of MTT. The results and figures show data on cytotoxicity obtained using the formula: Cytotoxity=(St-Sp)/(T-Sp)*100%, where St is the number of stained cells; Sp, spontaneously stained cells; T, total cells. In this formula control cell death and non-specific lysis are taken into account. We removed the mention of MTT from the Materials and methods chapter, and have added a formula for calculating the cytotoxic activity to this chapter to take into account the reviewer's wishes.

Line 162-164. The authors analysed 20 fractions, however Fig. 1A shows only 16 fractions. If the remaining samples were excluded the reason should be included in the text.

Fractions 17 to 20 contained a significant amount of salts and for this reason were not analyzed for the inhibition of cytotoxic activity and the composition of peptides using MALDI. We have corrected the manuscript to add information that we analyzed only the first 16 fractions for inhibition of cytotoxic activity.

Figure 1A. What does the fraction number 1 represent? It has a lower percentage of dead cells than fraction 11. Which control was used? Have the data been normalized to untreated cells? How many times was the experiment repeated? Statistics should be added.

In fraction 1, the exclusion volume of the column was eluted, i.e., proteins and peptides that exceeded its limit in molecular weight. MALDI analysis showed that this fraction contains a certain amount of non-hydrolyzed Tag7 protein. Since we previously demonstrated that the Tag7 protein itself is able to inhibit TNF-dependent cytotoxic activity (JBC 2015), we assumed that the effect of inhibiting cytotoxicity caused by 1 fraction is associated with the activity of the non-hydrolyzed Tag7 protein, and did not consider it further. This information is added to the Results Chapter. Untreated cells in the RPMI-1640 medium and the untreated cells with aliquots from the studied fraction added were used as controls. The cytotoxicity of the control cells ranged from 3% to 5%.

This experiment was repeated 4 times while we selected the best parameters for obtaining and separating the resulting peptides using chromatography. Figure 1 shows the profile for the last experiment. We added data on statistics to the Materials and methods and Figure legends, and changed the Figures according to the reviewer's suggestions.

Line 176-177. What is the rationale behind the nomenclature of the peptide? The fraction isolated was the number 11. The names 17.1 and 17.0 are somewhat confusing.

After obtaining a sequence of peptides that inhibit the cytotoxic activity of TNFa, using MALDI, we have ordered a chemical synthesis of homogeneous peptides for further work. Our manufacturer assigned numbers to the synthetic peptides, which we retained to refer to these peptides, which admittedly, in retrospect, could be confusing.

Figures 2C and 3B. The quality of the western blots is not acceptable.

We have repeated the experiments shown in Figures 2C and 3B and now are presenting a replicate new data.

Lines 210-212. Which concentration of TNFα did the authors use? If the increase in concentration of 17.1 peptide causes a decrease in cytotoxicity mediated by TNFα, is the opposite also true? It would be more informative to repeat the experiment in Figure 3B using an increasing concentration of TNFα.

The concentration of TNFa used in the experiments was 5*10-11M. We have added new data on the inhibition of cytotoxic activity depending on the concentration of TNFa. A sharp decrease in the inhibitory effect of peptide 17.1 concurrently with the increase in the TNFa concentration indicates a displacement of peptide 17.1 from the complex with the TNFR1 receptor. These data are quite informative, since they allow us to assess the quantitative characteristics of this interaction. To displace peptide 17.1 from a complex with the TNFR1 immobilized on Sepharose, a TNFa concentration 10-fold higher than the 17.1 concentration was required to block the inhibitory effect of peptide 17.1. According to the reviewer's suggestion, we have added information to Figure 2 on the restoration of the cytotoxic activity of TNFa by increasing its concentration and displacement of the inhibitory peptide.

Lines 236-240. The authors claim that Hsp70 interacts with the 17.1 peptide, and this is supposedly shown by western blot in Figure 4A. However the figure does not show what the authors claim but it is referred to in lines 241-246. The authors need to show the western blot they are referring to.

We deeply apologize to the reviewer since indeed we have not provided this Figure in our ms. We are grateful to the reviewer for pointing out this error. Figure 4A has been added to the ms.

Table 1. In this experiment it would have been appropriate to measure the release of inflammatory cytokine such as IL-6 to corroborate the conclusions.

In CFA-induced arthritis, the main inflammatory processes occur inside the synovial cavity. Measurements of the concentration of proinflammatory cytokines in the peripheral blood are not informative due to their low degree of infiltration. In addition, measuring the concentration of proinflammatory cytokines inside the joint in a mouse is not possible, due to the inability to obtain a sufficient amount of biomaterial (trace amounts of synovial fluid). This can serve as a goal for further work.

Round 2

Reviewer 1 Report

The manuscript was significantly improved during the revision and can be published now.

Reviewer 2 Report

The authors have responsed adequately to my comments and concerns.